# Hypernetwork Link Prediction Method Based on Fusion of Topology and Attribute Features

**DOI:** 10.3390/e25010089

**Published:** 2022-12-31

**Authors:** Yuyuan Ren, Hong Ma, Shuxin Liu, Kai Wang

**Affiliations:** 1People’s Liberation Army Strategic Support Force Information Engineering University, Zhengzhou 450001, China; 2National Digital Switching System Engineering and Technological R&D Center, Zhengzhou 450001, China

**Keywords:** attribute hypernetwork, link prediction, hypergraph learning, attention mechanism

## Abstract

Link prediction aims at predicting missing or potential links based on the known information of complex networks. Most existing methods focus on pairwise low-order relationships while ignoring the high-order interaction and the rich attribute information of entities in the actual network, leading to the low performance of the model in link prediction. To mine the cross-modality interactions between the high-order structure and attributes of the network, this paper proposes a hypernetwork link prediction method for fusion topology and attributes (TA-HLP). Firstly, a dual channel coder is employed for jointly learning the structural features and attribute features of nodes. In structural encoding, a node-level attention mechanism is designed to aggregate neighbor information to learn structural patterns effectively. In attribute encoding, the hypergraph is used to refine the attribute features. The high-order relationship between nodes and attributes is modeled based on the node-attribute-node feature update, which preserves the semantic information jointly reflected by nodes and attributes. Moreover, in the joint embedding, a hyperedge-level attention mechanism is introduced to capture nodes with different importance in the hyperedge. Extensive experiments on six data sets demonstrate that this method has achieved a more significant link prediction effect than the existing methods.

## 1. Introduction

Various networks can describe a large number of complex systems in the real world. Scholars generally use complex networks to describe real systems. A complex network consists of nodes and connected edges, in which nodes can be used to represent different individuals, and links between two nodes can represent their interactions. However, in many practical problems, these relationships may exist among multiple actors, not only including low-order pairwise relationships. For example, in the coauthorship network [1], multiple authors jointly publish a paper; in the metabolic network [2], multiple chemicals participate in a reaction. Obviously, in these cases, when modeling the properties of these systems with a single network consisting of paired nodes, there will be limitations to varying degrees, and much valuable information will be lost. Hypernetworks provide an effective way to model such high-order topological features. Hypernetworks are the generalization of networks. Different from the connection of pairs of nodes, hyperedges can represent the relationship between multiple nodes, so a hypernetwork can fully represent the higher-order correlation between entities. Scholars have paid extensive attention to hypernetworks in recent years owing to the efficiency and sufficiency of their modeling. For example, in social networks, hypernetworks can intuitively represent the interaction, and message transmission between friend groups, such as personalized recommendation [3]; in the knowledge graph, the hypernetwork can maintain multiple relationships between facts, such as knowledge base completion [4], intelligent question and answer [5], and other tasks. In addition to containing relational network topology information, attribute hypernetworks can also explicitly capture the attribute information of entities. For example, papers in citation networks also have attributes, such as abstract and title. The reasonable use of this cross-modal semantic information can enhance the analytical expression ability of the network.

As an important task in network science, link prediction aims to explore the formation mechanism and evolution law of the network to predict the connection edges that actually exist in the network but cannot be observed or not yet observed. It mines hidden pattern information in data through modeling association between objects, which is widely used in various fields, such as information retrieval of knowledge maps [6], interactive prediction of proteins [7], and accident assessment of traffic networks [8]. Some of the current methods focus on homogeneous information networks [9,10], learning the characteristics of the network from the two dimensions of topology and node attribute information for link prediction, while others focus on heterogeneous information networks [11,12], which takes into account the semantic relevance of different node types and connection relationships in the network compared with the former, and has received more extensive research. However, these methods all follow with interesting low-level semantic information about entities, ignoring the high-order complexity commonly existing in real networks, which has become the limitation of using traditional network modeling methods to solve link prediction tasks. For example, in the paper coauthorship network [1], a valuable scientific research achievement usually involves multiple authors. The traditional research method can only establish a link relationship based on whether the authors have cooperated with each other due to the constraint of the binary relationships. The specific and diversified cooperation information is ignored, including the participation relationship of multiple authors, the detailed attributes of all authors, and even other cooperative associations within the cooperative group. Similarly, in the metabolic network [2], a complete reaction must contain at least three chemical substances. Traditional methods can only establish a relationship based on whether two substances can react. As for the specific information of the reaction, including the attribute information of other participants, their mutual influence, and the inclusion relationship of different reaction groups are only lost by this single flattening mapping. It can be seen that these networks driven by higher-order relationships often contain semantic information from multiple perspectives. How to accurately capture potential semantic information to ensure the integrity of modeling has become the key to link prediction. The hyperlink prediction method can more accurately depict the interaction between entities and solves the problem of information loss in mapping multivariate relations to binary relations. Although many scholars have conducted extensive research on the variable cardinality problem [2,13], the tuple level measurement problem of hyperedges [14,15], and the capture problem of local and global structures [16,17] in the prediction of hypernetwork links, all of which measure the possibility of the existence of hyperedges in terms of the high order of topology, there is less research on how to use the complex interaction between entities and attributes to build deep network models. There is also a higher-order mapping relationship between entities and attributes. Multiple entities may have the same attributes. This potential association often contains the inherent relevance of the network structure. Using the information interaction between entities and attributes can help the model to carry out more expressive embedded learning. Although attribute hypernetwork can capture more abundant semantic information, it is a challenging task. How to embed the content attributes of entities in high-order and complex interactions and then mine the cross-modal associations between data is very essential for the comprehensive and accurate modeling of network features.

In response to these challenges, in this paper, we design a link prediction method oriented to attribute hypernetworks. This method learns both structural and attribute modal information and combines the attention mechanism for more targeted training, which effectively enhances the performance of link prediction. The contributions of this paper are as follows:We propose a general link prediction framework for the attribute hypernetwork, which integrates the structure and attribute characteristics of entities. Through a dual channel encoder, the interaction between network structure and node attributes is captured, and adopting hypergraph neural networks learns high-order interactions of nodes and attributes so that node information can be effectively transmitted through shared attributes.We introduce the node level-attention mechanism in structure encoding to model the importance of node neighbors to capture the neighborhood information of nodes; We introduce the hyperedge-level attention mechanism in the joint embedding module to consider the influence of different nodes in each hyperedge to screen the key nodes from a large amount of information. In this way, high-quality node embedding and hyperedge embedding can be obtained.Numerous experiments on six datasets have verified the efficiency of the TA-HLP model in link prediction. Compared with the latest hyperlink prediction model [2,10,14,18,19,20,21,22,23,24,25], the TA-HLP model has significantly improved under the AUC and recall indicators. At the same time, ablation experiments have also proved the effectiveness of the joint encode and the attention mechanisms.

## 2. Related Work

With the continuous breakthrough of artificial intelligence and network science, link prediction has been extensively studied in machine learning. Traditional link prediction is mainly based on similarity index or network embedding method to judge the probability of links between pairs of nodes. The former primarily uses the scores of static indicators to calculate the probability of link existence, such as Common Neighbor [26], Local Path [27], Katz [28], etc. The network embedding method learns the vector representation of nodes for link prediction tasks. DeepWalk [18], LINE [19], and Node Node2vec [10] define different sampling strategies to embed the obtained node sequence. Some emerging graph neural networks are also used in graph representation learning. They adopt different aggregation methods to map the original information of the network, such as graph convolution networks [29], graph self-encoders GAE [30], VGAE [31], and GraphSAGE [32]. However, these methods are all aimed at the network constructed by the common graph, and the reality often requires modeling between nodes that go beyond pairwise association. Therefore, our approach uses a hypergraph to show and represent the high-level interaction between multiple nodes to comprehensively and effectively capture the rich and diverse topology of the network.

Hypernetwork is an important extension and promotion of ordinary networks, which can more intuitively model the more complex high-order structure inside the system. At present, most research on hypernetworks focuses on embedded learning. The purpose of hypernetwork embedding is to learn a low dimensional embedding space, retain the information of the original network in the embedding space as much as possible, and screen out redundant information and noise information in the data so that the similarity of edge nodes embedded should be higher than that of irrelevant nodes. It is used to perform subsequent tasks such as classification, clustering, link prediction, subgraph segmentation. Recently, hypergraph neural networks [22] and hypergraph convolution networks(HyperGCN) [23] have been proposed to encode higher-order data correlations in hypernetwork structures. In addition, a large amount of research is used to explore the indecomposability of heterogeneous hypernetworks. Tu et al. [14] designed a deep embedding model DHNE to achieve the nondecomposable cluster similarity function. Huang et al. [33] learned a hyper-path-based random walk model HPHG to retain the structural information of the hypernetwork. Yu et al. [15] proposed a goal that is applicable to both uniform and non-uniform hypergraphs with nonnegative embedding vector constraints. In addition, the newly proposed embedding schemes [24,25,34] also focus on the self-learning problem of the hypernetwork. Compared with the fixed learning mechanism of the former, they define different attention modules to achieve dynamic information aggregatiotab, further improving the flexibility and effectiveness of the hypernetwork representation. However, most of these methods only measure the embedding of hyperedges from the topology of the network. Although some neural network methods [22,23,24,25] for hypergraphs consider the attributes of nodes, they only use attributes to initiate the node characteristics. They ultimately learn the vector representation of nodes based on topology, ignoring the deep interaction between entities and attributes, and being unable to fully integrate multivariate attribute information into the measurement of entity features. Therefore, our method captures the content attributes of entities while maintaining the high-order topology between entities and separately encodes the embedded representation of entities and attributes, thus, retaining the semantics jointly reflected by both to the greatest extent.

## 3. Notations and Problem Statement

Table 1 summarizes the commonly used symbols and definitions.

**Definition** **1**(Attribute Hypernetwork)**.**
*An attribute hypernetwork can be defined as G={V,E,X}, including a node set $V=\{v_{1},\cdots ,v_{m}\}$, a hyperedge set $E=\{e_{1},\cdots ,e_{n}\}$ and an attribute characteristic matrix of nodes $X={\left[x_{1},\ldots ,x_{m}\right]}^{T}$. Each $e_{i}$ is a set of partial nodes in* V*, and each node $v_{i}$ has a d-dimensional attribute vector $x_{i}$. $m=\left|V\right|$ represents the number of nodes, $n=\left|E\right|$ represents the number of hyperedges, and $d=\left|x_{i}\right|$ represents the dimension of node attributes.*

**Definition** **2**(Structure Incidence Matrix)**.**
*The topology of the hypernetwork G can be described by a structure incidence matrix $S\in R^{m\times n}$, and each element $S_{ij}$ in the matrix is defined as:*
(1)$$S_{ij}=S\left(v_{i},e_{j}\right)=\left\{\begin{array}{cc} 1, & v_{i}\in e_{j}\\ 0, & \;\mathrm{otherwise}\; \end{array}\right.$$

**Definition** **3**(Attribute Incidence Matrix)**.**
*Corresponding to the given node attribute characteristic matrix X, the relationship between nodes and attributes in the hypergraph can be expressed by an attribute incidence matrix $H\in R^{m\times d}$, and each element of the matrix $H_{ij}$ is defined as:*


(2)
$$H_{ij}=S\left(v_{i},a_{j}\right)=\left\{\begin{array}{cc} 1, & X_{ij}\neq 0\\ 0, & \;\mathrm{otherwise}\; \end{array}\right.$$


**Definition** **4**(Degree Matrix of Nodes and Attributes)**.**
*Given a node $v_{i}\in V$, its degree is defined as $d\left(v_{i}\right)=\sum_{a_{j}\in A}H\left(v_{i},a_{j}\right)$. For an attribute $a_{i}\in A$, its degree is defined as $d\left(a_{i}\right)=\sum_{v_{j}\in V}H\left(v_{j},a_{i}\right)$. According to this, we can get the degree matrix of nodes $D_{v}$ and the degree matrix of attributes $D_{a}$. Obviously, they are diagonal matrices. The diagonal elements are the degrees of nodes and attributes, respectively, and the rest are all 0. The definitions are as follows:*
(3)$$\begin{array}{cc} D_{v} & =\left(\begin{array}{cccc} d\left(v_{1}\right) & 0 & \cdots & 0\\ 0 & d\left(v_{2}\right) & \cdots & 0\\ \vdots & \vdots & \ddots & \vdots \\ 0 & 0 & \cdots & d\left(v_{n}\right) \end{array}\right) \end{array}$$
(4)$$\begin{array}{cc} D_{a} & =\left(\begin{array}{cccc} d\left(a_{1}\right) & 0 & \cdots & 0\\ 0 & d\left(a_{2}\right) & \cdots & 0\\ \vdots & \vdots & \ddots & \vdots \\ 0 & 0 & \cdots & d\left(a_{n}\right) \end{array}\right) \end{array}$$

**Problem** **1**(Attribute Hypernetwork Representation Learning)**.**
*The purpose of attribute hypernetwork representation learning is to map the nodes of the original network into a deep embedding in a low-dimensional space where the structural and attribute characteristics of the node are retained. Formally, given an attribute hypernetwork G={V,E,X}, it aims to learn a function $f:G\mapsto Z^{SE},Z^{AE}$ to map the nodes of the original network into potential vectors in low dimensional space. $Z^{SE}\in R^{m\times d_{1}}$ and $Z^{AE}\in R^{m\times d_{2}}$ are, respectively, the learned structure embedding and attribute embedding of nodes.*

**Problem** **2**(Hypernetwork Link Prediction)**.**
*The problem of hypernetwork link prediction is a tuple $\left(E,\bar{E}\right)$. E represents the hyperlink set of a known incomplete hypernetwork, and $\bar{E}=2^{V}-E$ represents a candidate set composed of unknown hyperlinks. The task of hypernetwork link prediction is to learn a hyperlink scoring function $s:e_{i}\mapsto y_{i}\in R$ so that the hyperlink allocation score in the known link set E is as high as possible than that in the unknown link set $\bar{E}$.*

## 4. Proposed Method

The proposed TA-HLP model is a deep link prediction model for the attribute hypernetwork. The framework is shown in Figure 1, which is composed of an encoder that jointly learns the structure and attributes of the hypernetwork and a decoder that performs hyperlink prediction. The central processing procedures are as follows:Based on the topological relationship, the structure encoder adopts a node-level attention mechanism to learn neighbors with different weights, so as to obtain the structural feature embedding of nodes.Based on attribute information, the attribute encoder uses hypergraph neural network to capture the potential association between entities and attributes to obtain the attribute feature embedding of nodes.The decoder combines the structure and attribute features in two independent semantic spaces, and then adopts the hyperedge-level attention mechanism to learn the importance of different nodes in each hyperedge. After the weight information is aggregated, the hyperedge is embedded.Input hyperedge embedding and learn a hyperlink scoring function to judge the hyperedges that have more remarkable similarity with the original network’s mode and behavior at the two levels of network structure and content attributes, i.e., the possible missing hyperedges of the original network.

### 4.1. Encode

#### 4.1.1. Structure Encoder

Structure encoder embeds entity structure information through feature mapping and aggregates node structure features based on weighted neighbors, effectively capturing the impact of different levels of neighborhood information on node embedding to improve the quality of node representation. The specific process is as follows:

First, we utilize a nonlinear transformation to embed different types of nodes from a heterogeneous space into a common potential space, input node feature X, and obtain the feature representation of nodes in the embedded space ${\tilde{Z}}^{\mathrm{SE}}$:(5)$${\tilde{Z}}^{\mathrm{SE}}=\sigma \left(XW^{SE}+b^{\mathrm{SE}}\right)$$

σ(•) represents the nonlinear activation function. $W^{SE}\in R^{d\times d^{SE}}$ and $b^{SE}\in R^{d^{SE}\times 1}$ represent the weight parameters and bias parameters to be learned for the nonlinear transformation layer respectively. Then, the node-level attention mechanism is employed to characterize the relative importance between nodes and their neighbors. The importance weight $\omega_{ij}$ of node $v_{j}$ to node $v_{i}$ is expressed as
(6)$$\omega_{ij}=\mathrm{attn}\left({\tilde{Z}}_{v_{i}}^{\mathrm{SE}},{\tilde{Z}}_{v_{j}}^{\mathrm{SE}}\right)=\sigma \left(a^{⊤}\cdot \left[{\tilde{Z}}_{v_{i}}^{\mathrm{SE}}∥{\tilde{Z}}_{v_{j}}^{SE}\right]\cdot W^{\mathrm{NA}}\right)$$

$W^{NA}\in R^{\frac{d_{1}}{2}\times d^{SE}}$ and $a\in R^{d_{1}\times 1}$ are the parameters to be trained for the attention layer. ∥ indicates the concatenate operation. After softmax normalization: (7)$$a_{ij}=\frac{\exp\left(\omega_{ij}\right)}{\sum_{v_{k}\in N_{i}}\exp\left(\omega_{ik}\right)}$$

Based on this importance weight, a weighted message transmission mechanism is adopted; that is, a new embedded representation of the node $Z_{v_{i}}^{SE}$ is obtained through the weighted sum of the feature vectors learned previously: (8)$$Z_{v_{i}}^{SE}=\sum_{v_{j}\in N_{i}}a_{ij}{\tilde{Z}}_{v_{j}}^{SE}$$

#### 4.1.2. Attribute Encoder

In this paper, a hypergraph is employed to model the higher-order relationship between nodes and attributes. Entities with the same attributes are mapped to a hyperedge. The incidence matrix $H\in R^{m\times d}$ can describe the many-to-many relationship between nodes and attributes. For any two entities, the more hyperedges associated with them, the more similarities will appear at the attribute level, and the closer they should be in the embedded feature space, which is consistent with the core idea of hypergraph embedding. Therefore, in the attribute encoding module, we use the transformation and aggregation mechanism of hypergraph neural network to re-encapsulate and represent entities and attributes, so that node information can be more spread through shared attributes. Compared with ordinary deep learning methods, we capture attribute features through hypernetworks, and entities and attributes can be updated bi-directionally using hyperedges, which is no longer just simple aggregation. Each layer of the hypergraph neural network can extract the potential associations of entities in attribute dimensions through convolution operation. Figure 2 shows the overall structure of the attribute encoder. The specific process is as follows:

The hypergraph neural network learns attribute embedding through aggregation node embedding. Input the node attribute matrix *X*, and obtain the attribute embedded $Y^{AE1}$ after linear transformation of a hyperedge convolution layer: (9)$$Y^{AE1}=H^{T}D_{v}^{-1/2}X^{AE1}$$

Similarly, hypergraph neural network updates node embedding through aggregation attribute embedding. According to the obtained attribute embedding $Y^{AE1}\in R^{d\times d_{2}}$, update the node embedding as $X^{AE2}\in R^{m\times d_{2}}$ after each hyperedge convolution. $W^{AE2}\in R^{d_{2}\times d_{2}}$ is the filter matrix to be learned by the second layer hypergraph convolution network. The updated formula is as follows: (10)$$Z^{AE2}=\sigma \left(D_{v}^{-1/2}HD_{a}^{-1}Y^{AE1}W^{AE2}\right)$$

The attribute encoder based on a hypergraph neural network superimposes a two-layer hypergraph convolution network. It updates the nodes and attributes by collecting node features and attribute features, finally obtaining the embedded representation $Z^{AE}$ of nodes on attribute features.

### 4.2. Decode

#### 4.2.1. Feature Fusion

Since the structure encoding module and attribute encoding module get the representation of nodes in two semantic spaces, before decoding, entities from different encoding modules are mapped into the same vector space through feature fusion. $Z^{SE}$ represents the embedded representation of the node on the structural characteristics, $Z^{AE}$ represents the embedded representation of the node on the attribute characteristics, and ∥ represents the concatenate operation. The final embedded representation of the node $Z^{V}$ is obtained through a linear transformation:(11)$$Z^{V}=W^{FF}\left(Z^{SE}∥Z^{AE}\right)$$

$W^{FF}\in R^{d_{v}\times \left(d_{1}+d_{2}\right)}$ is the parameter that the feature fusion module needs to learn.

#### 4.2.2. Hyperedge Embedding

By fusing the outputs of the two encoders, the embedded representation of each node contains semantic information of different dimensions, which makes the nodes that make up the hyperedge have different structures and attributes, resulting in a large difference in their impact on the hyperedge. Therefore, it is necessary to design a hyperedge-level attention module to mine the importance of different nodes, so as to aggregate multi-source information selectively and adaptively determine the most valuable node in the hypergraph propagation process. Specifically, first, we obtained the activation vector of node $v_{i}$ through a nonlinear function, then calculated its product with the learning parameter *w*, and finally standardized the product through a softmax layer. The formula for calculating the importance weight of node $v_{i}$ is as follows: (12)$$a_{i}=\sigma_{1}{\left(w^{⊤}\sigma_{2}\left(W^{EA}Z_{i}^{v}\right)\right)}^{⊤}$$

$\sigma_{1}\left(•\right)$ and $\sigma_{2}\left(•\right)$ represent different nonlinear activation functions, and $W^{EA}$ and *w* are learning parameters with the size of $d^{EA}\times d_{v}$ and $d^{EA}\times 1$, respectively. Finally, the embedded representation of the hyperedge is obtained by the weighted sum of the embedded vector of the node and its weight coefficient: (13)$$Z^{e}=\sum_{i\in e}a_{i}Z_{i}^{v}$$

#### 4.2.3. Hyperlink Prediction

The purpose of the decoder is to learn a hyperlink scoring function to calculate the similarity of each hyperlink and screen the hyperlinks that best match the original network. The definition is as follows: (14)$$s\left(e\right)=\sigma \left(W^{SC}Z^{e}+b^{SC}\right)$$

$W^{SC}\in R^{d_{e}\times d_{v}}$ and $b^{SC}\in R^{d_{e}\times 1}$ are the training parameters of the hyperlink prediction layer.

### 4.3. Loss Function

$E_{1}$ represents observed hyperlinks, that is, known relationships between vertices, $2^{V}-E_{1}$ represents a group of unknown relationships, including some actual but unobserved hyperlinks, and $E_{2}$ represents a group of sampling sets from unknown relationships. The purpose of optimization is to maximize the number of hyperlinks scoring higher than the average score of $E_{2}$ in $E_{1}$, that is, to make the observed hyperlinks scoring higher than the unobserved hyperlinks as much as possible. The definition is as follows: (15)$$L=\frac{1}{|E_{1}|}\sum_{e_{i}\in E_{1}}\ell \left(\frac{1}{|E_{2}|}\sum_{e_{j}\in E_{2}}S\left(e_{j}\right)-S\left(e_{i}\right)\right)$$

ℓ(•) represents a non-decreasing function ℓ(x)=log(1+exp(x)), and functionis used here. For each known hyperedge $e_{i}\in E_{1}$, we generate a corresponding hyperedge $e_{j}\in E_{2}$. One-half of the nodes in $e_{j}$ come from the random sampling of $e_{i}$, and the other half is from the random sampling of the remaining node set $V-e_{i}$. To avoid experimental deviation caused by hyperlink size, we ensure that the number of nodes of each known hyperedge $e_{i}$ and the corresponding unknown hyperedge is the same $e_{j}$.

Algorithm 1 summarizes the optimization learning process of the TA-HLP model.
**Algorithm 1** TA-HLP.**Input:**$S\in R^{m\times n}$: Structure incidence matrix, $H\in R^{m\times d}$: Attribute incidence matrix, *X*: Attribute characteristic matrix, *T*: Max epochs**Output:**$Z^{v}\in R^{m\times d_{v}}$: Embedded representation of nodes, $Z^{e}\in R^{n\times d_{e}}$: Embedded representation of hyperedges1: **for** epoch = 1 to *T* **do**2:   Implement semantic feature mapping of nodes via Equation (Equation 5)3:   Calculate the importance weight of the neighborhood of the nodes via Equation (Equation 7)4:   Adopt weighted message passing mechanism to update the structure embedding of nodes via Equation (Equation 8)5:   Use hypergraph neural network to update the attribute embedding of nodes via Equation(Equation 10)6:   Combine structure and attribute features to obtain the final embedding of nodes via Equation(Equation 11)7:   Learn the importance weight of different nodes in the same hyperedge via Equation (Equation 12)8:   Aggregating node information with different weights to obtain embedding of hyperedge via Equation (Equation 13)9:   Construct a scoring function to predict the missing hyperlink via Equation (Equation 14)10:   Minimize loss function (Equation 15) to update model parameters11: **end for**

## 5. Experiment

### 5.1. Datasets

This paper uses real hypernetwork datasets to test the prediction ability of the designed model, including two coauthored network datasets: CoauthorshipCora (represented by CC) and CoauthorshipDBLP (represented by CD), two citation network datasets: SocietationCiteseer (represented by CCI) and SocietationPubmed (represented by CP), one organic response network dataset: OrganicReaction (represented by OR), and one restaurant access dataset: YelpRestaurant (represented as YR). The statistical scale of the data set is shown in Table 2, which is described in detail as follows:

Coauthorship network [1,35]: Coauthorship datasets provide a way to analyze research collaboration. The author is represented by nodes, and the cooperation relationship participating in the same paper is characterized by hyperedges. The purpose is to anticipate other potential collaborations.

Citation network [35]: the paper is represented by nodes, and the features of each paper are the word bag representation of the document. In this paper, we generate hyperedges based on the hyperedge extension strategy proposed in the paper [22]. The purpose is to speculate on unknown reference relationships.

Organic reaction network [2]: The reconstruction of a chemical reaction network is a fundamental problem in computational chemistry. Chemical substances are represented by nodes, and the reaction relationship between organic substances is represented by hyperedges. The basic properties of substances constitute the characteristics of nodes. The purpose is to reconstruct a complete reaction network.

Restaurant access network [36]: The restaurant is represented by a node, and the hyperedge is formed by selecting the restaurant visited by the same user. Node features are composed of a hotspot code of latitude, longitude, city, and state, and the word codes of the first 1000 words in the corresponding restaurant name. The purpose is to recommend the favorite restaurant groups of users.

### 5.2. Baseline Methods

This paper selects the mainstream network embedding method and hyperlink prediction method as the baseline method.

DeepWalk [18]: This method obtains the node sequence by truncated random walk and then uses word2vec to transform the nodes into low-dimensional vectors.

LINE [19]: This method defines the first-order and second-order similarity of nodes and then minimizes the KL divergence representing the similarity and the actual similarity to obtain the vector representation of nodes.

node2vec [10]: In this method, the random walk strategy is weighed by parameters during the sampling process to generate high-quality node representation.

The above methods [10,18,19] does not provide a computer system for hyperlink similarity. This paper uses the following scoring function to measure the difference between hyperlinks: (16)$$S\left(e\right)=\sigma \left(\frac{1}{\left|e\right|}\sum_{v_{i}\in e,v_{j}\in e,i\neq j}x_{i}^{T}x_{j}\right)$$
|e| is the number of nodes in the hyperedge, and $x_{i}$ is the learned node representation.

CMM [2]: This method alternates the nonnegative matrix decomposition and least square matching in the adjacent space of vertices to find the most suitable hyperlink for incomplete networks.

C3MM [20]: This method introduces a “clique closure” hypothesis based on the CMM method to improve the performance of the algorithm.

DHNE [14]: The method defines a nonlinear bundle similarity function to keep the nondecomposability of the hypernetwork and designs a deep automatic encoder to capture the structure of the hyper network.

NHP [21]: NHP-U is a neural hyperlink predictor that uses GCN to refine the embedding of each vertex.

HGNN [22]: This method uses the hypergraph convolution operation to learn the hidden layer representation of higher-order data structures to code the hypergraph data correlation.

HyperGCN [23]: The nonlinear Laplacian operator is introduced to define the GCN on the hypergraph.

Hyper-SAGNN [24]: This method proposes a new self-attention mechanism to solve the representation learning problem of heterogeneous hypergraphs.

Hyper-Atten [25]: This method adds an attention module to the incidence matrix of the hypergraph to dynamically update the internal relationship between vertices.

Since the above methods [22,23,24,25,25] do not perform the hyperlink prediction task, the hyperlink scoring function proposed in this paper is used for hyperlink prediction.

### 5.3. Evaluation Indicators

To comprehensively and accurately evaluate the prediction ability of the algorithm, this paper selects the common link prediction indicators AUC score (the Area Under the receiver operating characteristic Curve) and recall. AUC is a commonly used index to measure link prediction accuracy as a whole. It is defined as follows: (17)$$\begin{array}{c} \mathrm{AUC}=\frac{\sum_{i=1}^{N}\sum_{j=1}^{m}I\left(\;\mathrm{pred}\;\left(x_{i}\right),\;\mathrm{pred}\;\left(y_{j}\right)\right)}{N^{*}M}\\ I=\left\{\begin{array}{cc} 1, & a>b\\ 0.5, & a=b|\\ 0, & a<b \end{array}\right. \end{array}$$
pred(•) is the prediction score of the model for the genuinely missing hyperlink and the non-existent hyperlink. N and M represent the numbers of two types of samples, respectively.

The recall is an indicator to measure whether the algorithm prediction is comprehensive, that is, how many missing hyperlinks in the network are predicted. Here, we rank the prediction scores from high to low and select the prediction results of the first *k* (*k* is half of the actual number of missing) to calculate the Recall@k (represented by R@k), which is defined as follows: (18)$$R@k=\frac{n}{L}$$
*n* is the correct predicted result in the top k rankings, and *L* is the actual number of missing hyperlinks in the network.

### 5.4. Experimental Design and Analysis

In the experiment, we randomly hide 20% of the hyperedges are randomly hidden as the test set, 70% of the hyperedges and the randomly generated unknown hyperedges are employed for model training, and 10% of the hyperedges are used as the verification set to optimize the hyper-parameters. In the training process, we use the Adam algorithm [37] to optimize our loss function and set the learning rate to 0.001. In the hyperedge-level attention module, $\sigma_{1}\left(•\right)$ selects the softmax function, $\sigma_{2}\left(•\right)$ selects the tanh function, and the rest of the nonlinear activation functions σ(•) select ReLU. Set the same embedding dimension $d_{1}=d_{2}=128$ in both the structure and attribute encoder and use dropout to avoid overfitting, with a value of 0.5. All parameters in the algorithm are optimized through a grid search strategy. To eliminate the influence of random errors and evaluate the experimental results as accurately as possible, 100 independent experiments were carried out on each network, and the average results were obtained, as shown in Table 3. The AUC score and R@k of the TA-HLP model and eight benchmark models are analyzed below.

From the overall observation, compared with all baseline methods, TA-HLP has achieved the most significant effect on all indicators under different datasets. The AUC score increased by 5–10 percentage points, and the recall increased by 5–7 percentage points. The superiority of the model in link prediction performance is verified. TA-HLP can enhance the expression learning ability of the original network and achieve more comprehensive and accurate prediction.DeepWalk, LINE, and node2vec are all traditional complex network embedding methods. To predict hyperlinks, we calculate the average value of pairwise similarity to represent the similarity of the entire hyperedge, neither maintaining the irresolvability of the hyperedge nor ignoring the content attributes of the node. Although CMM and DHNE use hypergraphs to model the relationship between nodes that transcend pairwise connections, they ignore the attribute information of nodes. Therefore, the performance of these two methods on data sets is poor, which proves that both higher-order structure and attribute modalities are necessary for link prediction. Their complementary relationship can enhance the depth of network embedding and better characterize network information. C3MM believes that the newly generated hyperlinks in the network are more likely to have evolved from the recently closed cliques, mainly from the perspective of temporal evolution. Similarly, there is no systematic learning for the attribute information of data, so our data is only slightly improved compared with the CMM method.NHP, HGNN, HyperGCN, Hyper-SAGNN, and Hyper-Atten all combine the deep neural network, and also consider the network structure and attribute characteristics. The experimental results also prove their effectiveness. However, in the modeling process, they simply use the attribute characteristics of the convolutional layer aggregation node. They do not fully use the potential higher-order interaction between entities and attributes, resulting in poor prediction performance. To fully mine the association between nodes and attributes, this model adopts a hypergraph structure to model the correlation between nodes and attributes. Each attribute is defined as a hyperedge and nodes with the same attribute form the adjacency relationship. At the same time, the deep association between nodes and attributes is captured based on the “node attribute node” transformation. We also introduce a two-layer attention mechanism for reinforcement learning of important nodes, so the experimental results of TA-HLP are obviously better than other algorithms.In addition, the Hyper-Atten model has the best performance among all baseline methods. This may be because the Hyper-Atten model additionally applies an attention module to the incidence matrix to update the degree of association between nodes in real-time. It can be seen that more attention to important content is effective. However, compared with the Hyper-Atten model, our model can distinguish the importance of different levels in a more fine-grained way. It measures the impact of different components on network embedding from the perspectives of nodes and hyperedges, respectively, so that the embedded representation of the network can more truly restore the connotation of structure and semantics, which is also an important reason for the significant improvement of our model performance.

### 5.5. Ablation Experiment

To further illustrate the effectiveness of the joint encoding module and the two attention mechanisms in the TA-HLP model, we conducted the following ablation experiments. This paper designs four variants of the algorithm. TA-HLP-1 removes the structure coding module from the original model, and TA-HLP-2 removes the attribute coding module from the original model. Both variants only learn the structure or attribute characteristics of nodes. Compared with TA-HLP itself, these two methods show poor performance. This shows that compared with a single learning module, the dual channel encoder proposed in this paper can more effectively learn the vector representation of entities. They capture the similarity of entities from different dimensions, retain more diversified semantic information in the original network, and have obvious advantages in improving link prediction performance. TA-HLP-3 removes the node-level attention mechanism and only studies the importance of each node for hyperedge embedding. TA-HLP-4 removes the hyperedge level attention mechanism and considers that the nodes in the same hyperedge have the same degree of influence and only consider the contribution of node neighbors to the structure pattern. Figure 3 shows the experimental results proving the effectiveness of the two attention mechanisms. The prediction performance of the two variants on each data is lower than that of the proposed algorithm, which proves the advantages of the two attention strategies in the encoding and decoding modules. They can help the model extract more critical information, carry out targeted learning of node embedding and hyperedge embedding, and, thus, improve the expression ability of the algorithm.

### 5.6. Attention Mechanism Analysis

When learning embedding vectors of hyperedges, we consider the importance of different nodes and assign them different weights. To better understand the meaning of weight, we conduct an in-depth analysis of the hyperedge-level attention mechanism through a specific case. CoauthorshipDBLP is a network dataset of co-authored papers, which records the publication of papers from four different research fields. For details, refer to Section 5.1 of the article. Taking hyperedge as an example, Figure 4 shows the topological relationship of $e_{543}$, and Figure 5 shows the attention weights of different nodes in the embedded representation of $e_{543}$. $e_{543}$ is a paper published by a machine in the field of learning. It includes four authors $v_{1013}$, $v_{2152}$, $v_{3203}$, and $v_{29019}$, respectively. The weight of the first three nodes is larger, and the weight of the last node is smaller. According to the bag feature vector of the paper, the authors represented by $v_{1013}$, $v_{2152}$, and $v_{3203}$ have published a large number of papers in the field of machine learning, and they also have cooperative relations with other papers in the field, so they have more similarities with the category and affiliation, and make greater contributions to the publication of the paper $e_{543}$. However, most of the papers published by $v_{29019}$ are in the field of information retrieval, which is just an extension of his research field. Therefore, the features of $v_{29019}$ cannot well express the information of hyperedge $e_{543}$. $v_{29019}$ is not a key node to form hyperedge embedding and should be given a lower weight. According to the above analysis, the hyperedge-level attention weight can evaluate the contribution of nodes to the hyperedge, and assign higher weights to some valuable nodes.

### 5.7. Parameter Sensitivity Analysis

To analyze the impact of different parameters on the TA-HLP model, we conducted experiments on node embedding vectors of different dimensions on six datasets. The results are shown in Figure 6. In general, before the TA-HLP model achieves the best effect, with the increase in embedding dimension, more and more node features are captured by the encoder, and the prediction results of the model are steadily promoted; however, when the optimal performance is reached, too many embedded dimensions may lead to information redundancy, which will not only increase the computational burden of the model but also interfere with the fitting ability of the model, resulting in a continuous decline in prediction performance. When the embedding dimension is 64 or 128, the TA-HLP model achieves the best effect. When the dimensions are 8 and 1024, the impact of the model is poor. It can be seen that too few or too many embedding dimensions may result in insufficient or invalid information used by the model, which may lead to overfitting and poor effect.

## 6. Conclusions and Future Work

In this paper, we study the problem of link prediction in attribute hypernetworks. Aiming at how to collect more semantic information from the original network for comprehensive and multivariate reconstruction, a general framework is proposed to learn the network structure and attribute information jointly. The encoder consists of a structure encoder based on node level attention mechanism and an attribute encoder based on the hypergraph neural network. It extracts entity features from the topological structure and attributes content, respectively, fully uses the semantic association between nodes and attributes, and captures the cross-modal interaction of higher-order structure and node content. In addition, the hyperedge-level attention mechanism is introduced in the decoder to aggregate nodes with different contributions in the hyperedge. The experimental results also demonstrate that the proposed method can deeply mine the composite interaction characteristics of the network and effectively improve the link prediction performance of the model. This shows that hypernetworks can not only effectively model high-order mappings between entities and relationships, but also accurately depict interactions between entities and attributes, which further expands the research direction of hypernetworks and is a very valuable work for in-depth exploration of the issue of “why the complex system has complexity”.

In the future, it is worth further exploring how to design an embedding method that integrates attribute-level attention mechanisms. At the same time, extending the proposed method to temporal networks to perceive the evolution pattern of structure and attribute characteristics is also an important research topic of attribute hypernetwork prediction.

## Figures and Tables

**Figure 1 entropy-25-00089-f001:**
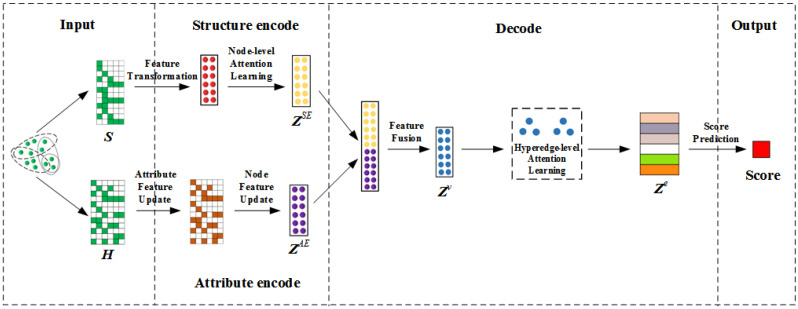
The framework of hypernetwork link prediction method for fusion topology and attributes (TA-HLP).

**Figure 2 entropy-25-00089-f002:**
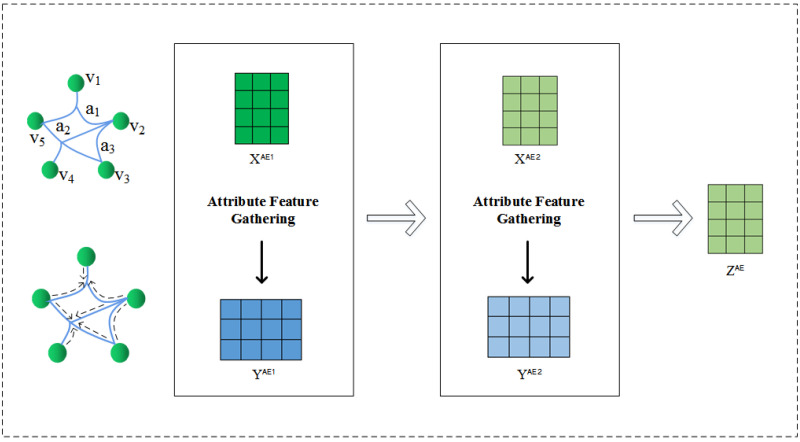
Overall framework of attribute encoder based on hypergraph neural network.

**Figure 3 entropy-25-00089-f003:**
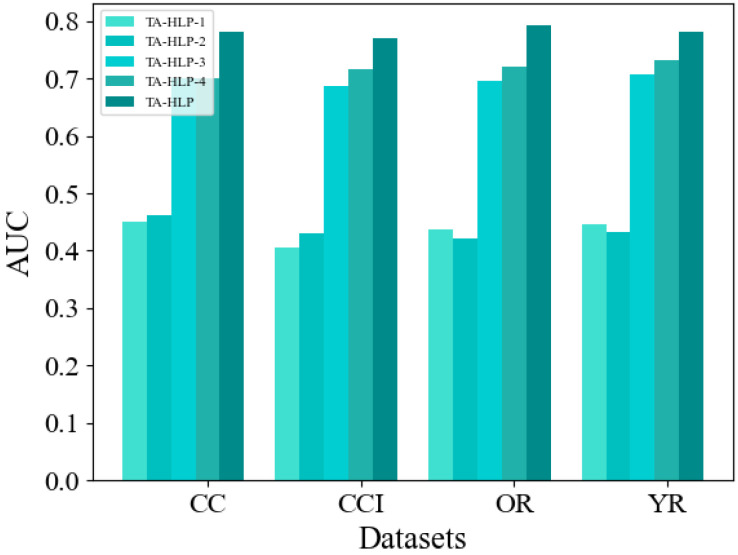
AUC score results of ablation study.

**Figure 4 entropy-25-00089-f004:**
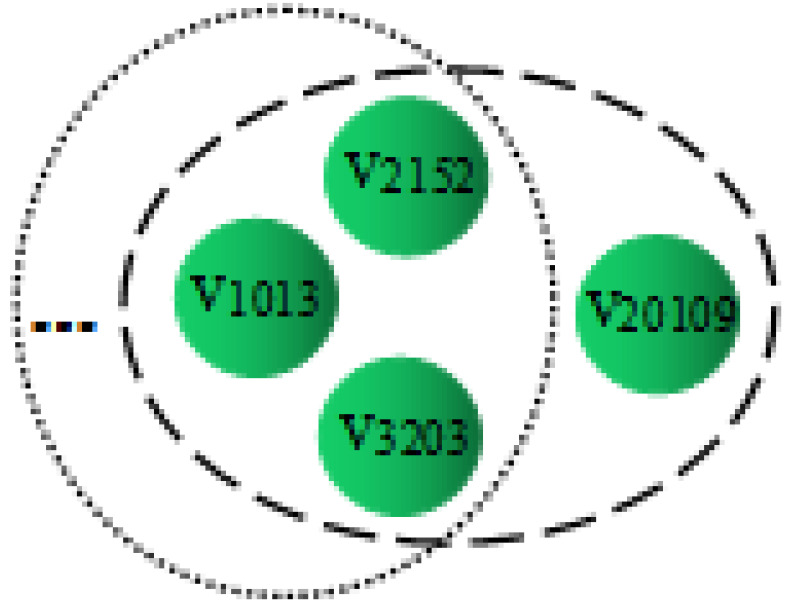
Topology of $e_{543}$.

**Figure 5 entropy-25-00089-f005:**
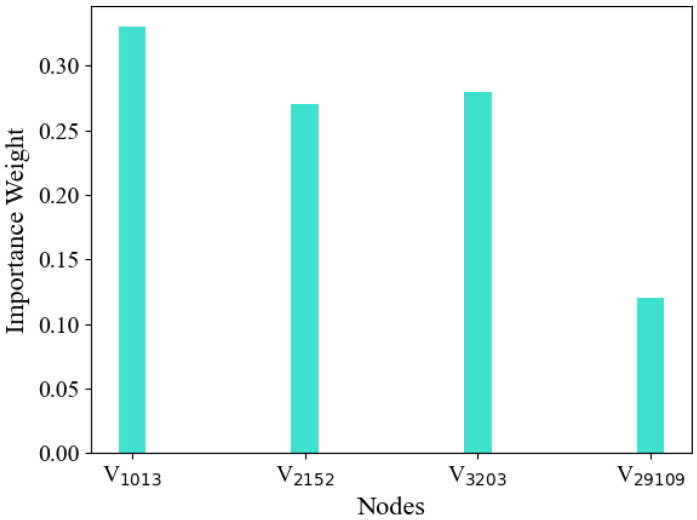
Attention weight of component nodes of $e_{543}$.

**Figure 6 entropy-25-00089-f006:**
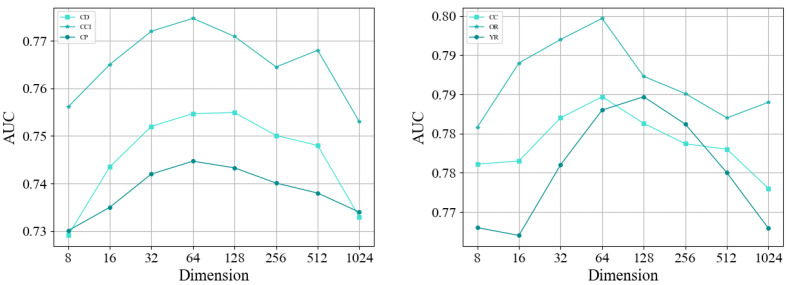
Impact of different dimensions of embedded vectors for hyperlink prediction.

**Table 1 entropy-25-00089-t001:** Notations.

Notation	Description
G={V,E,X}	Attribute hypernetwork.
*V*	A set of nodes.
*E*	A set of hyperedges.
*X*	An attribute characteristic matrix.
*m*	The number of nodes.
*n*	The number of hyperedges.
*d*	The dimension of node attributes.
$S\in R^{m\times n}$	Structure incidence matrix.
$H\in R^{m\times d}$	Attribute incidence matrix.
$Z^{SE}\in R^{m\times d_{1}}$	Structural embedding of nodes.
$Z^{AE}\in R^{m\times d_{2}}$	Attribute embedding of nodes.
$Z^{v}\in R^{m\times d_{v}}$	Embedding of nodes after fusion.
$Z^{e}\in R^{n\times d_{e}}$	Embedding of hyperedges.

**Table 2 entropy-25-00089-t002:** Comparison of datasets’ sizes.

	Node	Hyperedge	Attribute
CC	2708	1072	1433
CD	41,302	22,363	1425
CCI	3327	1079	3703
CP	19,717	7963	500
OR	16,293	11,433	298
YR	50,758	679,302	1862

**Table 3 entropy-25-00089-t003:** Comparison of AUC score and R@k.

	CC		CD		CCI		CP		OR		YR	
	AUC	R@k	AUC	R@k	AUC	R@k	AUC	R@k	AUC	R@k	AUC	R@k
DeepWalk	0.53	0.23	0.57	0.23	0.54	0.21	0.54	0.24	0.56	0.26	0.55	0.26
LINE	0.56	0.25	0.52	0.25	0.55	0.24	0.58	0.22	0.61	0.24	0.54	0.25
Node2vec	0.55	0.25	0.58	0.25	0.58	0.23	0.62	0.26	0.57	0.27	0.58	0.28
CMM	0.64	0.22	0.67	0.27	0.64	0.31	0.63	0.29	0.66	0.33	0.68	0.29
C3MM	0.66	0.26	0.68	0.26	0.64	0.32	0.67	0.32	0.68	0.27	0.71	0.32
DHNE	0.62	0.29	0.65	0.26	0.61	0.31	0.63	0.30	0.62	0.31	0.64	0.30
NHP-U	0.69	0.28	0.68	0.29	0.71	0.32	0.68	0.31	0.70	0.29	0.73	0.33
HGNN	0.68	0.27	0.69	0.23	0.69	0.29	0.69	0.32	0.71	0.27	0.71	0.33
HyperGCN	0.66	0.28	0.70	0.28	0.67	0.30	0.70	0.31	0.69	0.31	0.66	0.32
Hyper-SAGNN	0.63	0.26	0.68	0.24	0.64	0.28	0.68	0.30	0.65	0.28	0.65	0.31
Hyper-Atten	0.68	0.30	0.70	0.31	0.71	0.34	0.70	0.31	0.71	0.32	0.73	0.35
TA-HLP	**0.78**	0.35	0.75	0.34	0.77	0.37	0.74	0.39	0.79	0.39	0.78	0.38

## Data Availability

The data presented in this study are available on request from the corresponding author.

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
