# Peer review of "Hypernetwork Link Prediction Method Based on Fusion of Topology and Attribute Features"

_entropy, 2022, doi:10.3390/e25010089_

Round 1
Reviewer 1 Report
Summary
The authors propose an approach for link prediction in hypernetworks with attributes. Both structural information and attribute information is taken into account and combined in a model using attention. The approach is evaluated on common benchmarking datasets and an ablation study is performed.
Pros
- Model outperforms baselines
- Interesting model using both structural information and attribute information.
- Nice ablation study investigating the effects of different parts of the models
Cons
- The related work and baselines appear to be outdated. The best baseline model HyperGCN is from 2018, NHP from 2019. However, there are newer works, for example:
- Zhang, R., Y. Zou, and J. Ma. "Hyper-SAGNN: a self-attention based graph neural network for hypergraphs." International Conference on Learning Representations (ICLR). 2020.
- Bai, Song, Feihu Zhang, and Philip HS Torr. "Hypergraph convolution and hypergraph attention." Pattern Recognition 110 (2021): 107637.
- How was the hyperparameter optimization done? It seems that there was no separate validation set which might lead to overfitting on the test set.
Details
- Clearly define hypernetworks in the introduction. Doing this in the related work section is a bit late: "Hypernetworks are the generalization of networks in which hyperedges can represent the relationship between multiple nodes."
- What exactly is the difference between a hyperlink and a hyperedge. This should be stated more clearly.
- The matrix D_v is not defined.
- Table 3: reaches into the right margin of the page. Reduce the space between columns or choose shorter column headings?
- The conclusion only summarizes the article but does not discuss the impact of this work.
Reviewer 2 Report
Please see the attachment.

Reviewer 3 Report
comments:
* presentation needs to be improved. The figures are in low resolution, some text is not readable.
* Table 3, right part is cut out of the page.
* Figure 2, why split into two figures?
* reproducibility is low: the authors did not provide how to reproduce the result
* what is the threshold for recall? it is important but not reported
* many minor things (definition 4 repeated, equation 14 |e|, etc.)
* Equation 4, what is || "splicing operation"? Please use a more common term or explain it in detail.
Round 2
Reviewer 2 Report
Weaknesses-
Major weaknesses :
** Include a figure to represent the architecture of the hypergraph neural network developed.
A graphical representation of how hyperedge-level attention mechanisms are performed with an example of joint embedding using one of the real world 6 datasets is required for better understanding of the mechanism.
Line 18: “Complex networks provide a unique research idea for depicting various relationships 18 in the real world.” Instead of this definition, give a more meaningful introduction to what a complex network is.
Line 23: Give proper references to “coauthorship network” and “metabolic network”.
Line 24: “Obviously, in this case,” should be as “Obviously, in these cases,”, which refers to the practical problems included in the previous sentence.
Line 24: what does a “single network” refer to? Is it a simple network which is the counterpart of a complex network?
Line 30 :
Paraphrase this sentence to convey the meaning well.
“In recent years, because of the efficiency and sufficiency of its modeling, hypernetworks have received extensive attention from scholars.”
Ex: “Scholars have paid extensive attention to hypernetworks in recent years owing to the efficiency and sufficiency of its modeling.”
Line 55: What is “paper coauthorship network”? Is it a paper ? Give proper citations.
Line 58: Grammatical error-binary relationship should be as binary relationships.
Line 61: Give proper citations to metabolic networks.
Line 139: what is ​​aggregatiotab3n?
Line 294, Line 298, Line 302, Line 307 - Give proper citations for these datasets.
Line 357 - Is Recall@k in line 357 and R@k in equation 18 the same or different?
Line 345, “Because the above methods does not ” better replace with “Since the above methods do not”
Line 545 - Ambuiguity in reference. What does “ref8” means?
Minor weaknesses
Line 354- Recall, R to be simple letter
Line 355 - Should be as “Here, we rank the”, missing comma
Line 411 - “But compared with…” comma should be after “But,”
Round 3
Reviewer 2 Report
Authors have addressed the previous issues in the review processes.